# Breaking the Bottleneck in Anticancer Drug Development: Efficient Utilization of Synthetic Biology

**DOI:** 10.3390/molecules27217480

**Published:** 2022-11-02

**Authors:** Haibo Wang, Yu He, Meiling Jian, Xingang Fu, Yuheng Cheng, Yujia He, Jun Fang, Lin Li, Dan Zhang

**Affiliations:** 1Department of Laboratory Medicine, Sichuan Provincial People’s Hospital, School of Medicine, University of Electronic Science and Technology of China, Chengdu 610072, China; 2Sichuan Provincial Key Laboratory for Human Disease Gene Study, Sichuan Provincial People’s Hospital, School of Medicine, University of Electronic Science and Technology of China, Chengdu 610072, China

**Keywords:** natural product, cancer therapy, molecular mechanism, synthetic biology

## Abstract

Natural products have multifarious bioactivities against bacteria, fungi, viruses, cancers and other diseases due to their diverse structures. Nearly 65% of anticancer drugs are natural products or their derivatives. Thus, natural products play significant roles in clinical cancer therapy. With the development of biosynthetic technologies, an increasing number of natural products have been discovered and developed as candidates for clinical cancer therapy. Here, we aim to summarize the anticancer natural products approved from 1950 to 2021 and discuss their molecular mechanisms. We also describe the available synthetic biology tools and highlight their applications in the development of natural products.

## 1. Introduction

Cancer is one of the deadliest diseases, in which cell grow uncontrollably and then are shed into the blood, which transports the cells to other parts of the body. Cancer has been highlighted as one of the most serious health challenges worldwide. According to global cancer statistics, in 2020, 19.3 million new cancer cases and 10.0 million cancer deaths were reported [1]. In particular, in 2020, the cumulative risk of cancer incidence and mortality increased to 20.58% and 10.73%, which are 2.08 and 1.90 times higher than these in 2002, respectively [2]. Thus, an increasing number of researchers are focusing on cancer research and therapy [3,4]. Conventional cancer therapies include surgery, radiation, and chemotherapy. Clinical cancer drugs play essential roles in cancer therapy. However, chemotherapy is often accompanied by the emergence of resistance, cancer recurrence and the development of serious side effects, such as peripheral neuropathy, nephropathy and liver injury [5,6]. Therefore, there remains an urgent demand for clinical drugs with excellent curative effects against cancer.

Natural products (NPs) originate from animals, plant extracts, insects, marine organisms, microorganisms and other organisms. They are produced by a series of mechanisms in hosts, mainly including polyketide synthase (PKS), nonribosomal peptide synthetase (NRPS) and ribosomally synthesized and posttranslationally modified peptide (RiPP) mechanisms. Due to the wide variety of biosynthetic mechanisms, NPs exhibit diverse structures, biological activities and applications, especially in the treatment of human diseases and in veterinary and agricultural applications [7]. As primary and secondary metabolites or intermediates, NPs do not involve dangerous chemical synthesis processes and byproducts.

NPs, such as the anticancer drug paclitaxel [8], immunosuppressive drug rapamycin [9] and antiparasitic drug artemisinin [10], have been extensively applied for the prevention and treatment of various human diseases. As of 2019, 64.4% of the 1881 drugs approved by the United States Food and Drug Administration (U.S. FDA) are NPs or their derivatives and analogues, such as plant-derived artemisinin and microorganism-derived fidaxomicin [11]. At present, nearly 70% of antibacterial drugs and 65% of antitumor drugs come from NPs and their derivatives [12], such as daptomycin [13] and actinomycin. Due to their structural diversity, NPs have a broad array of anticancer mechanisms. They can inhibit cancer development by arresting cell proliferation and promoting cell apoptosis, autophagy and immunotherapy via a series of signaling pathways [14,15].

To discover novel and valuable NPs, an increasing number of strategies and tools have been developed and applied. Synthetic biology is the most popular and powerful strategy for discovering novel and valuable NPs. As an emerging field of biological research in the early 21st century, synthetic biology aims to artificially design and construct new biological systems with specific physiological functions to establish biological manufacturing pathways for products, such as drugs, functional materials or energy substitutes. To design and construct biological systems, numerous tools and methods have been developed, such as the antibiotics and secondary metabolite analysis shell (antiSMASH) web server [16], clustered regularly interspaced short palindromic repeats-CRISPR associated protein (CRISPR-Cas) [17,18], DNA assembly [19], direct cloning [20], transcriptional regulation [21], protein engineering [22], multiple omics [23] and host engineering [24].

In this review, we will summarize the approved anticancer natural drugs covering from 1950 to 2021 and discuss the great potential of the clinical applications of NPs in cancer therapy and summarize their anticancer mechanisms. We will also outline the synthetic biology strategies and tools used for discovering valuable NPs and highlight their applications in NP development.

## 2. FDA-Approved Anticancer Natural Products

As a typical and general medical therapy for cancer, chemotherapeutic drugs can also damage normal cells and then cause a series of complications. Moreover, drug resistance emerges with the wide use of chemotherapeutic drugs, especially in advanced-stage cancers. All these factors can result in the failure of cancer therapy. Thus, there is a need for drugs with strong activities against cancer cells and low toxicity against normal cells. Different NPs, on the one hand, exhibit effective activities against various types of cancer; on the other hand, they play prominent roles in drug resistance via various mechanisms [25,26,27]. Moreover, they can also improve the effects of immune therapy [15]. In brief, natural products are becoming significant for medicine research.

In this section, we summarize all FDA-approved anticancer NPs, covering 72 years from 1950 to 2021. In total, 87 NPs are summarized and discussed according to their names and sources (Table 1). Forty-five percent of anticancer drugs are derived from large biological macromolecules, NPs or their derivatives (Figure 1A). Specifically, the number of NPs applied in cancer therapy has grown rapidly from 1950 to 2020 (Figure 1B, Table 1). In the 1950s, 7 NPs were approved for application in clinical cancer therapy. Meanwhile, in the 2010s, 21 NPs were approved for cancer therapy, three times the number approved in the 1950s. The anticancer cancer NPs approved in the last two decades account for 41% of all approved NPs (Figure 1C). This suggests that with the development of technology, an increasing number of NPs are being found and applied in clinical cancer therapy. More interestingly, the number of steroids decreased gradually, and only one hormone was approved in the last decade, for therapy of castration-resistant prostate cancer, which is four times lower than the number approved in the 1950s (Figure 1D, Table 1). On the other hand, marine NPs and antibody drug conjugates (ADCs) have emerged and increased rapidly in the last two decades, and they will be the main sources of clinical drugs. Except for these approved NPs, a large number of NPs have been reported to be toxic to cancer cells through various mechanisms, and some of them entered clinical trials [28,29,30]. These unapproved NPs still have great potential as clinical drugs.

## 3. Anticancer Mechanisms of Natural Products

As we know, human cancer is produced from multiple processes. The cancer cells in different processes seem to exhibit different functional capabilities. Hanahan D. envisages eight distinct hallmark capabilities, including evading growth suppress, avoiding immune destruction, enabling replicative immortality, activating invasion and metastasis, inducing or accessing vasculature and so on, to rationalize the complex phenotypes of diverse human tumor types and variants [118]. Destroying the hallmark capabilities in tumor progresses is promising for tumor suppression.

NPs suppress cancer development via diverse anticancer mechanisms due to their various chemical structures (Figure 2). For example, the intact lactone ring of camptothecins and the colchicine domain of podophyllotoxins are widely recognized as the active structures for their anticancer activities [119,120]. Although the antitumor effects of some NPs have been reviewed previously [14,121], the molecular mechanisms of NPs in cancer development have not been comprehensively summarized and discussed. Thus, we further summarize the molecular mechanisms of NPs in tumor suppression in this section.

### 3.1. Plant-Derived Drugs

Most natural anticancer drugs identified to date are plant-derived NPs, mainly derived from paclitaxel, camptothecin, podophyllotoxin, vinca-alkaloid and their analogues. Due to the numerous types of plant-derived drugs, their anticancer mechanisms are varied.

Microtubules play significant roles in the proliferation, migration and invasion of cancer cells. Various anticancer agents have been designed to target microtubules. There are two effective antimicrotubule therapeutics, paclitaxel and vinca-alkaloids. Paclitaxel and its analogues bind to tubulin and then promote tubulin polymerization and inhibit microtubule disassembly [122]. In contrast to paclitaxel, vinca-alkaloids and their analogues, such as vinflunine, vinblastine and vincristine, inhibit tubulin polymerization and subsequent spindle assembly [123] (Figure 3A). Microtubule dynamics change dramatically over time, which is essential for promoting microtubule growth and disassembly. Both of these classes of drugs induce the failure of microtubule dynamics. Microtubule dynamics instability is highly responsible for mitotic chromosome instability, which is common in cancer cells.

Camptothecin analogues, including belotecan HCl, topotecan HCl and irinotecan HCl, kill cancer cells by specifically targeting and binding to DNA topoisomerase I, which results in a double-stranded DNA break during the S-phase [124]. Another class of NPs, podophyllotoxin analogues, have been identified as topoisomerase II inhibitors. The most recent reports suggest that etoposide and its metabolites interact with topoisomerase II enzymes on the one hand, and they can also covalently bind to CREB-binding protein and T cell protein tyrosine phosphatase on the other hand [54]. These two proteins are functional in the differentiation and proliferation of hematopoietic stem cells. The binding of etoposide directly results in enzyme inhibition. DNA topoisomerases are essential in DNA replication, transcription and chromosome segregation. Their inhibitors block these biological processes and then disrupt cell proliferation.

In addition, some other plant NPs, such as homoharringtonine, arglabin and solamargines, suppress tumor growth via specific pathways (Figure 3B). For example, arglabin and solamargines induce apoptosis via the mitochondrial pathway, which is one of the major mechanisms of apoptosis leading to programmed cell death. In this pathway, upon stimulation by apoptosis signal(s), cytochrome c is released from mitochondria to the cytoplasm and then interacts with Apaf-1 to form apoptotic bodies. The structurally changed Apaf-1 in the apoptotic body recruits caspase 9 and causes caspase 9 to be cleaved into two segments and activated. Activated caspase 9 further activates subsequent caspase proteins and then initiates apoptosis. Arglabin and solamargines inhibit the activity of Bcl-2 and upregulate caspase-3, respectively [125,126].

**Figure 3 molecules-27-07480-f003:**
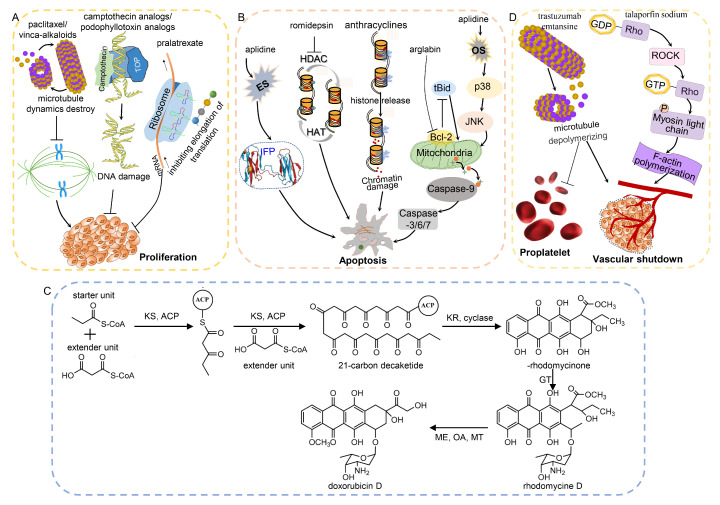
The main molecular mechanisms and regulation networks of anticancer drugs derived from natural products. (**A**) Natural products inhibit cell proliferation by destroying the microtubule dynamic and DNA damage and inhibiting translation. (**B**) Natural products induce apoptosis via mitochondrial pathway and chromatin damage and by destroying the balance of HAT and HDAC. (**C**) Biosynthesis of doxorubicin, a type II polyketide [127]. ACP: acyl carrier protein. KS: ketosynthase. KR: ketoreductase. GT: glycosyltransferase. MT: methyl transferase. OA: oxygenase. ME: methyl esterase. (**D**) Natural products result in vascular shutdown via Rho pathway. ES: endoplasmic reticulum stress. OS: oxidative stress. HAT: histone acetylase. HDAC: histone deacetylase. IFP: incorrectly folded protein. TOP: topoisomerase.

### 3.2. Microorganism-Derived Drugs

*Streptomyces* strains are the main sources of microorganism-derived drugs. Although the genetic manipulation and fermentation of *Streptomyces* are time-consuming, these organisms possess rich precursors, cofactors and posttranslational modifications, which are essential for the diverse bioactivities of natural products [24]. Due to the diverse structures of microorganism-derived drugs, their mechanisms vary.

Nearly half of microorganism-derived drugs are anthracyclines, including aclarubicin, daunomycin, doxorubicin and their derivatives. They are synthesized via type II polyketide pathways [127] (Figure 3C). As the most effective curative regimen for a series of hematologic cancers and solid cancers, there are two important mechanisms by which doxymycin, daunomycin, epirubicin and idarubicin induce cancer cell apoptosis: DNA damage and chromatin damage [128] (Figure 3B). On the one hand, anthracyclines are composed of tetracycline aglycones related to aminoglycosides. The amino sugar inserts into the DNA minor groove, while the tetracyclic moiety inserts into the DNA double helix [129]. This action induces topoisomerase II poisoning and subsequent DNA double-strand breaks (DSBs). On the other hand, anthracyclines release histones from the genome, which results in chromatin damage and epigenomic and transcriptional effects. However, the combination of DNA and chromatin damage contributes to serious side effects, such as cardiovascular toxicity [130] and infertility [131]. Furthermore, as an analogue of doxymycin, aclarubicin evicts histones from the genome without inducing DSBs. It shows effective anticancer activity but less toxicity than other anthracyclines [131]. These results provide strategies with which researchers can improve the activity and safety of anthracycline drugs.

Additionally, bleomycin and mitomycin C preferentially cleave actively transcribed genes, and telomeric DNA is their major target. In contrast, bleomycin produces free radicals that act on chromosomes and then induce chromosomal aberrations [132]. Romidepsin is produced from by *Chromobacterium violaceum* and disrupts the balance of histone deacetylase by inhibiting the activity of class I histone deacetylase enzymes in refractory or relapsed cutaneous and peripheral T cell lymphomas (Figure 3B). Transcriptional suppression is another anticancer mechanism of microorganism-derived drugs, such as actinomycin D and mithramycin. Actinomycin D is an RNA polymerase inhibitor, and mithramycin is an inhibitor of Sp-1, which is a transcription factor of ACVRL-1 [75]. Temsirolimus is a derivative of sirolimus produced by *Streptomyces hygroscopicus*. It is used for the therapy of renal cell carcinoma by acting as an inhibitor of mTOR and subsequently inducing the autophagy of cancer cells [133].

### 3.3. Natural Product-Based ADCs

Cancer immunotherapies, including chimeric antigen receptor T cells, immune checkpoint blocks and vaccines, have been developed as promising treatments and have made significant therapeutic progresses in recent years. In contrast to traditional therapies, cancer immunotherapies have high accuracy, specificity and wide adaptability and significantly improve patient survival rate [134]. Monoclonal antibodies target immune checkpoints, which has revolutionized cancer treatment. Monoclonal antibodies can be used alone as immune checkpoint inhibitors. Additionally, they can be conjugated with potent cytotoxic agents and then deliver the cytotoxic agents precisely to the tumor cells.

Trastuzumab emtansine is a human epidermal growth factor receptor 2-targeted monoclonal antibody conjugated to emtansine. Brentuximab vedotin is an anti-CD30 antibody conjugated to monomethyl auristatin E. Both of these drugs improve cancer therapy efficacy by inhibiting microtubule generation via disrupting the polymerization of tubulin [135,136]. Both inotuzumab ozogamicin and gemtuzumab ozogamicin consist of a monoclonal antibody and a cytotoxic calicheamicin that bind DNA and subsequently result in DNA breaks. The former is directly delivered to refractory or relapsed acute lymphoblastic leukemia cells by anti-22 antibody and then induces DNA damage [137]. The latter is directly delivered to acute myeloid leukemia cells by anti-CD33 antibody. In general, to a certain extent, ADCs reduce the toxicity of drugs to normal cells due to their specificity and exhibit greater potential in cancer treatment than traditional therapy. However, autoimmune response and side effects still exist, and treatment cost is high, all of which hinder their widespread usage. Future developments in ADCs will focus on identifying better targets, more effective cytotoxic payloads and better linkers to improve the potency and safety of drugs.

### 3.4. Marine Natural Products

Compared with terrestrial plants, nonmarine microorganisms and animals, marine organisms possess greater potential to produce novel and bioactive NPs with diverse structures. They are considered to be the most recent sources of medicinal drugs. As of 2021, four marine NPs have been approved for cancer therapy.

The first marine anticancer drug to be approved was eribulin, which binds tubulin and induces microtubule depolymerization in breast cancer and soft-tissue sarcoma [138]. In addition, there are two marine drugs targeting transcriptional regulation, trabectedin and lurbinectedin. Trabectedin is produced from the marine animal *Ecteinascidia turbinate*. It is used to treat specific soft tissue sarcomas, such as liposarcoma and leiomyosarcoma. According to a clinical study, treatment with trabectedin produces better efficacy and a longer survival time in soft-tissue sarcoma patients [139]. Aplidine is produced from the marine microorganism *Aplidium albicans*, and it has curative effects in acute lymphoblastic leukemia via oxidative stress-mediated JNK and p38 activation and ER stress-mediated incorrect protein folding, triggering rapid apoptosis in cancer cells [140]. Lurbinectedin is a synthetic alkaloid related to trabectedin. As a new second-line therapy option, it was approved in 2021 for the treatment of metastatic small cell lung cancer. Lurbinectedin acts as a transcription inhibitor by binding to the minor groove of DNA and subsequently driving the accumulation of DNA breaks. Additionally, it can disrupt the interactions between DNA and protein and thereby disrupt RNA transcription [141].

With the development of techniques for sample collection and spectrometry, significant achievements have been made in the exploration of marine natural products. More than 32,900 marine NPs had been identified by 2020 [142,143,144,145]. The quantity of marine NPs is relatively high compared to that of synthetic compounds. Thus, their clinical trials are very promising.

### 3.5. Hormones and Other Natural Products

Hormones and their analogues suppress cancer development mainly by regulating androgen or estrogen levels. For example, formestane and testolactone function as aromatase inhibitors and decrease the estrogen levels of breast cancer and unresectable desmoid tumors, respectively [146,147]. Abiraterone acetate decreases the androgen level by acting as an inhibitor of CYP17 [148]. However, only one hormone analogue has been approved for cancer therapy. It seems that hormones are becoming less widespread in clinical use.

Moreover, some other NPs can also inhibit cancer development by modulating the immune system. Purine analogues and peptide analogues, such as cladribine and mifamurtide, have been shown to be active in a variety of B- and T cell malignancies [149,150]. They can activate immune cells, inducing an effective immune response and resulting in protection against cancer. Some cancer cells, such as multiple myeloma cells, are sensitive to the inhibition of the ubiquitin proteasome pathway induced by carfilzomib, since proteasome inhibition leads to the accumulation of unfolded proteins, endoplasmic reticulum stress and subsequent cell growth arrest [151]. As a modified epoxyketone, carfilzomib can bind to the functional sites of the 20S proteasome, inhibiting cell proliferation [152].

Photodynamic therapy is another attractive treatment technology. Unlike standard laser photocoagulation, photodynamic therapy enhances the antitumor effects by inducing vascular shutdown [153,154]. Vascular shutdown serves as a promising target for anticancer agents that bind to tubulin protein and lead to microtubule depolymerization [155] (Figure 3C). Similarly, talaporfin sodium evokes vascular shutdown via the RhoA/ROCK pathway, which directly mediates F-actin polymerization and subsequently destroys tumor vessels [110,156] (Figure 3C). The trastuzumab emtansine-induced disruption of microtubule dynamics summarized above can also block the formation of proplatelets and blood vessels [135].

## 4. Synthetic Biology Strategies and Tools for Discovering Natural Products

Given the great potential of NPs in clinical applications, an increasing number of pharmaceutical researchers have focused on NPs. Traditional methods are the primary approaches for discovering NPs because they involve the direct collection of compounds with biological activities from numerous organisms instead of genetic manipulation. Although traditional methods have been successfully applied in identifying many bioactive NPs, they have some limitations, such as being time-consuming, labor-consuming, having poor efficiency and requiring repeated extraction of known compounds. With the rapid development of sequencing, many genome sequences, RNA sequences and protein sequences have been shared in databases, such as the National Center for Biotechnology Information (NCBI) databases and UniProt. All these sequences light the avenues of bioinformatics, genetic manipulation, protein engineering and subsequent synthetic biology.

### 4.1. Bioinformatics Analysis

Databases provide vast information for researchers. For example, PubMed contains more than 26 million biomedical publications, which cover life science, chemical science, biological engineering, physical science and so on [157]. Based on the literature, researchers can extract target information that provides new insights for the therapeutic discovery and improvement of NPs against cancers. In addition, a series of genome analysis, proteome analysis and pathway analysis tools have been developed to discover hidden information. For example, antiSMASH is updated frequently and is used to predict and analyze the biosynthetic gene clusters (BGCs) of secondary metabolites in bacterial, fungal and plant genome sequences [16,158,159,160,161] (Figure 4A). These tools are constantly being extended and improved to help researchers identify unique and valuable metabolites.

High-throughput sequencing technologies are widely applied in genomics, transcriptomics, proteomics, metabolomics, lipidomics and single-cell sequencing (Figure 4B). These omics studies fill the knowledge gap regarding how many silent BGCs will yield novel NPs and then break through the bottleneck in BGC characterization [162]. Multiomics analysis provides crucial clues for revealing new insights into the biosynthetic mechanisms of NPs by integrating multilayer molecular information [163,164]. On the other hand, with the development of multiomics technologies and computer programs, there are enough genome sequences, RNA sequences, protein sequences and metabolites to be used for computational modeling and the subsequent generation of predictive biological systems and even metabolic reconstructions [165]. In a computational metabolic model, researchers can obtain a simulation that resembles a real laboratory experiment and an approximate outcome of the experiment by running a model in which researchers can adjust the parameters of every factor in the biosynthetic pathway to optimize the output of the end product (Figure 4C). For example, Michael C et al. achieved a three-fold improvement in the resveratrol titer in *Escherichia coli* (*E. coli*) by constructing a new probabilistic computational model [166]. In addition to metabolic modeling, computational models are also used to design cell factories to realize the optimal production of target molecules [167].

### 4.2. Pathway Reconstruction or Engineering

However, 90% of the NPs are in the dark, as they are produced at only low levels or not at all due to their silent BGCs. The rapid development of DNA sequencing technology has stimulated genetic manipulation, which is crucial for identifying and engineering novel and valuable NPs. Multiple-step biosynthetic pathways are known to be involved in NP biosynthesis, and these pathways contain diverse genes and their control elements. Researchers will be better able to discover these high-hanging fruits if these genes and their elements are optimally assembled into operational pathway(s). Therefore, new DNA assembly and engineering tools are arising constantly.

Pathway construction through DNA assembly in vivo or in vitro remains a fast and efficient method [168] (Figure 5A). Type II restriction enzyme-directed assembly of multiple DNA fragments in vitro and its derived technologies have various advantages due to their short cycle and convenient operations. On the other hand, the assembly of multiple overlapping DNA fragments based on homologous recombination in vitro or in vivo, as with Gibson Assembly, In-Fusion Snap Assembly and other assembly kits, is also popular in the laboratory because it can achieve seamless assembly. It can even realize a seamless construct of an entire bacterial genome (total size of 583 kb) or even a 900 kb DNA product with high efficiency [169]. However, errors may be introduced during in vitro recombination. Thus, homologous recombination in vivo, such as in yeast and *E. coli*, is powerful. It has not only high efficiency but also a low error rate. Additionally, pathway construction by direct cloning has gradually developed and matured (Figure 5B). The classic and widely used direct cloning method is genome library construction, which is completed by digesting the genome using restriction endonucleases and then ligating the genome fragments into expression vectors using T4. Bacterial artificial chromosome (BAC) libraries and transformation-associated recombination (TAR) are commonly used for NP discovery in the laboratory [170]. However, these methods are aimless, time-consuming and labor-intensive. Cas9-assisted targeting of chromosome segments (CATCH), a cloning technology that is designed based on CIRSPR-Cas9-mediated targeting digestion and Gibson assembly-mediated ligation, makes up for the shortcomings of seamless assembly [171]. This approach can be used to clone target genomic DNA sequences with a size of up to 100 kb. Another technology, RecET direct cloning, bypasses library construction and screening and directly clones large genomic DNA fragments into expression vector via RecET-mediated homologous recombination and Redαβ-mediated recombination [20,172].

Pathway engineering via single base or module editing based on CRISPR systems is a powerful strategy for discovering NPs and their derivatives and for improving NP titers (Figure 5C). Limited enzyme activity and poor protein stability are bottlenecks in the biosynthesis of NPs. Thus, researchers have focused on random or site-directed mutagenesis to improve enzyme activity and protein stability. Chen H et al. increased the titer of lycopene in Saccharomyces cerevisiae by 2.53 times by improving the catalytic activity of isopentenyl diphosphate isomerase via random mutation [173]. CRISPR/Cas-mediated base editing, an efficient technology for precise single-base editing of the genome in vivo, can precisely convert a C:G base pair to a T:A base pair by ligating a cytidine deaminase to a dead Cas (dCas) protein or convert an A:T base pair to a G:C base pair by ligating an adenosine deaminase to a dCas protein [174,175,176]. Gene deletion or insertion is another highly efficient method for NP discovery and titer improvement. The deletion of some negative regulators, competitive pathways or negative feedback pathways can lead to significant accumulation of an NP [177,178,179]. On the other hand, increasing the expression levels of key synthetic enzymes, rate-limiting enzymes or positive transcription regulators can also prominently improve NP production [180,181,182]. The production of spinosad showed a 1000-fold increase compared to its original production after overexpressing rate-limiting proteins by introducing strong promoters upstream of rate-limiting genes [183]. Among the NP biosynthetic enzymes, the sequences in modular NRPS or PKS show high similarities. Thus, engineering NRPS or PKS modular enzymes, including repeating the target functional modules and replacing a functional module with another functional or replacing the module with a heterologous module, will provide a platform to generate almost any desired derivative. Kudo K et al. obtained 19 rapamycin derivatives by editing modular polyketide synthase genes of rapamycin. CRISPR-Cas9, CRISPR-Cas12a and their derived tools are widely used to accurately edit single nucleotides, modules, genes or gene clusters [183].

### 4.3. Cell Factory

Host organisms are critical in the heterologous production of NPs because their metabolic efflux greatly influences NP production directly. Generally, natural producers are imperfect hosts for the production of NPs due to the lack of sufficient precursors or efficient genetic engineering techniques, slow growth rates or environmental perturbations. Thus, heterologous hosts are needed to ensure the highly efficient expression of NPs.

The cell factory, a self-replicating minimal cell, provides an optimal metabolic background for the biosynthesis of NPs. In the cell factory, genome mining is generally performed (Figure 6A). The nonessential genes or gene clusters are deleted from the genome to reduce the number of competing pathways; at the same time, extra integration sites are inserted into the genome to allow additional integration of multiple copies of a heterologous gene cluster [184,185]. Additionally, genetic regulation is an effective approach to optimize the cell factory. A series of regulators have been characterized and proven to be functional in NP biosynthesis [177,186]. Moreover, promoter engineering [187], transcription factor engineering [188], synthetic RNA switch and CRISPR-Cas systems [189] are also widely used to regulate gene expression and subsequent NP production in cell factories. Protein engineering has also become an increasingly popular method to improve protein activity, increase protein stability, maximize carbon flux through limiting steps, expand NP spectra and improve NP production [22]. A more complex issue is that the cell factory should be able to synthesize its own essential nutrients instead of acquiring them from the environment [190]. Therefore, understanding the minimum requirements and designing the cell factory from scratch are necessary. Metabolism engineering, including primary and secondary metabolism engineering, significantly increases the product pathway flux of target products due to the accumulation or balance of metabolic precursors, cofactors and energy. Metabolism engineering provides an optimal metabolic background for heterologous expression.

Microorganisms are the most popular cell factories, as they are easy to culture and genetically manipulate. *E. coli* and *Saccharomyces cerevisiae* (*S. cerevisiae*) are widely used due to their fast growth rates, easy genetic manipulations and high productivities [24]. Moreover, as the second family in which NPs were discovered, the Streptomycetaceae family has attracted much attention for its prolific drugs [191]. *Streptomyces* strains, especially *Streptomyces coelicolor* and *Streptomyces albus* [184], have rich cofactors and precursors. They are widely reengineered to construct efficient cell factories. For example, Myronovskyi M et al. constructed a cluster-free *Streptomyces albus* chassis strain by deleting these unessential gene clusters and introducing an additional phage *phiC31* attB site to increase the copy number of heterologous gene clusters. In the last, the authors significantly improved the production of six compounds including pyridinopyrone A, aloesaponarin II and didesmethyl-mensacarcin [184]. Moreover, *E. coli* and yeast strains have already been metabolically redesigned and engineered to overproduce intermediates of paclitaxel [192,193,194]. Although microorganisms exhibit many advantages, they have some limitations. For example, not all plant NPs, animal NPs and marine NPs can be expressed in microorganism hosts, as they lack efficient biosynthetic pathways or enzymes. To overcome these limitations, more and more plant cell factories [195], cyanobacteria cell factories and other cell factories have been developed. These heterologous plant or marine hosts provide alternative and sustainable avenues for the production of NPs that are not suitable for expression in microbial hosts.

### 4.4. Artificial Intelligence

With the rapid development of artificial intelligence, robotics and integrated software are also widely used to automate standard workflows, in which researchers write programs and robots complete the experiments according to the programs (Figure 6B). This workflow significantly improves the efficiency of synthetic biology and subsequently improves the speed of researchers to explore and create active compounds. Synthetic biology is a type of “on-demand manufacturing” composed of a variety of nonlinear workflows. However, the heterogeneity and dynamics of biological experiments make the workflows more complex due to the differences among batches [196]. In recent years, biological foundries have been developed with the integration of the automatic design–build–test engineering cycle, which solves several difficult problems in the processes of bioengineering, such as slow speed, high expense and low repeatability [197]. Mohammad H. et al. reported an application of an integrated robot system combined with a machine learning algorithm; this application completes the design, construction, testing and learning process of a fully automated biological system [198]. The fully automated robotic platform IOAutomata evaluated slightly less than 1% of the possible variation, which was 77% higher than that achieved by random screening. The authors also successfully used the robotic platform to optimize the biosynthesis of lycopene.

## 5. Conclusions and Future Perspectives

NPs have great potential in clinical cancer treatment due to their diverse structures and bioactivities. An increasing number of NPs, especially marine drugs and ADCs, have been applied in cancer treatment. With the emerging demand for novel natural drugs, effective methods for discovering NPs are needed. Although traditional approaches have been successfully applied to identify a large number of bioactive NPs, these approaches remain limited, as they are time-consuming, labor-intensive, “low-hanging” and inconsistent. With the conspicuous progress in genetic manipulation approaches and detection technologies, NP discovery and production improvement have become more promising.

Synthetic biology greatly promotes the exploitation of NPs. From traditional sample collection to complex genetic manipulation, and then to artificial intelligence, the exploitation rates and quantity of NPs are gradually increasing. For example, complex and powerful genetic manipulation greatly optimizes biosynthetic pathways and metabolic networks and provides an optimal background for target NP biosynthesis. Moreover, with synthetic biology, known compounds can also be modified through targeted design to obtain analogues with better activity and less toxicity or even new compounds. Specifically, through the application of artificial intelligence, more and more fully automated and multifunctional robotic platforms have been built and integrated. These systems overcome the above limits in the processes of bioengineering and greatly improve their efficiency and accuracy. Thus, efforts in synthetic biology will help NP discovery continue to thrive in the future.

Among the approved anticancer NPs, marine and antibody-conjugated NPs have rapidly emerged in the last decade (Figure 1D). These 32,900 marine natural products are only the tip of the iceberg. The organisms in the deep oceans have undergone the longest evolutionary period, and the harsh marine conditions have endowed them with a wide range of unique molecules. Thus, the potential of marine NPs in future drug discovery and clinical cancer treatment is immeasurable.

Additionally, smart drug delivery systems show strong competitiveness in cancer treatment. Smart delivery can improve the permeability, stability and solubility of drugs, promote the continuous controlled release of drugs and increase the therapeutic effects. In these smart delivery systems, the NPs are precisely delivered to target cancer cells, resulting in high toxicity to cancer and reduced side effects on the human body [199]. Although various drug delivery strategies have been explored to deliver chemotherapeutic drugs more directly to tumors, the most important transformation progresses have appeared in the field of ADCs. As an immune therapy, NP-derived ADCs represent novel biological systems for target treatment. Moreover, some NPs directly interact with and activate immune cells or pathways, often inducing protection against cancer [15]. Therefore, NPs also realize their clinical significance via immunotherapy in cancer treatment.

Furthermore, the complexity of refractory diseases has greatly weakened the therapeutic potential of existing treatment schemes. Therefore, clinical treatment has changed from monotherapy to combination therapy, which is now becoming accepted as an effective way to optimize efficacy while minimizing adverse effects [200]. Polatuzumab vedotin was first approved for the treatment of hematological malignancies, and combination therapy with rituximab produces a more significant efficacy than monotherapy in diffuse large B-cell lymphoma [201]. In addition to chemotherapy, polatuzumab vedotin can also be used with immunomodulatory therapy, which is promising for relapsed/refractory follicular lymphoma [202].

In summary, considering their various structures and bioactivities, the potential of NPs in clinical cancer treatment is highly promising. Specifically, emerging marine NPs and ADCs provide more choices of medications with higher efficacy and safety. Furthermore, combination therapy not only increases therapeutic efficacy but also helps to address drug resistance [203]. With the innovation of various technologies and popularization of artificial intelligence, a large number of NPs are being brought to our sights, providing more candidate clinical medications.

## Figures and Tables

**Figure 1 molecules-27-07480-f001:**
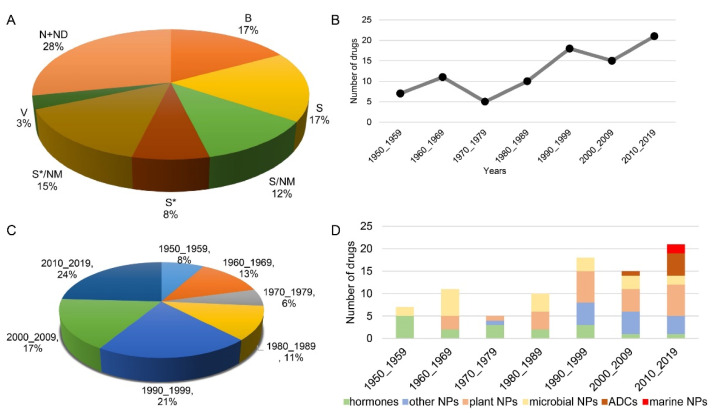
All approved anticancer drugs derived from natural products from 1950 to 2019. (**A**) The sources of anticancer drugs approved from 1950_2019 and their proportions. N: natural product. ND: natural product derivative. B: biological macromolecule. S: synthetic drug. NM: mimic of natural product. S*: synthetic drug with NP pharmacophore. V: vaccine. (**B**,**C**) Anticancer drugs derived from natural products. (**D**) All approved anticancer natural products by source/year. ADC: antibody drug conjugate.

**Figure 2 molecules-27-07480-f002:**
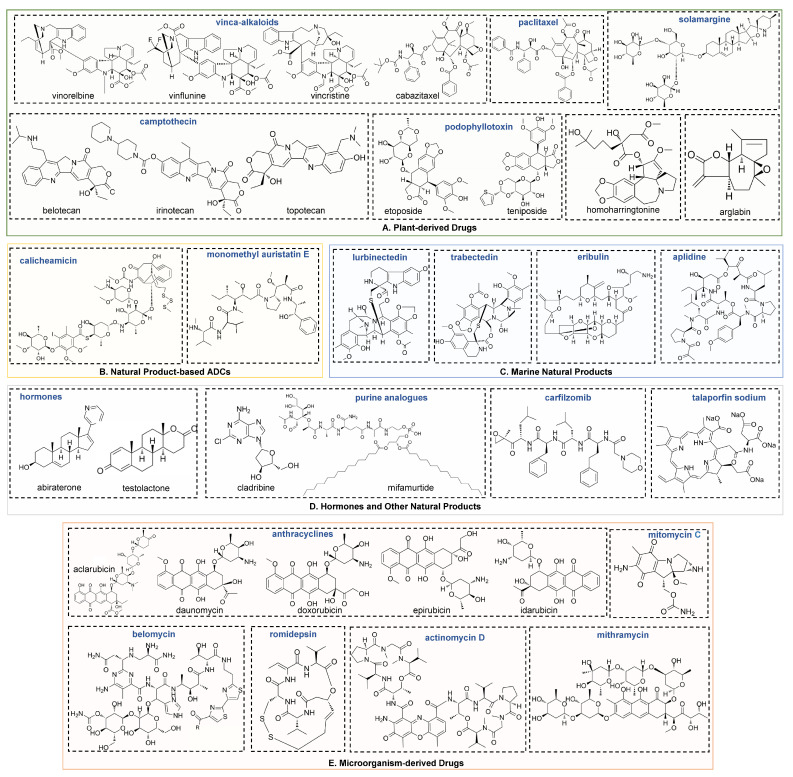
Chemical structures for plant-derived (**A**), ADCs (**B**), marine (**C**), hormones and other natural products (**D**) and microorganism-derived natural products (**E**).

**Figure 4 molecules-27-07480-f004:**
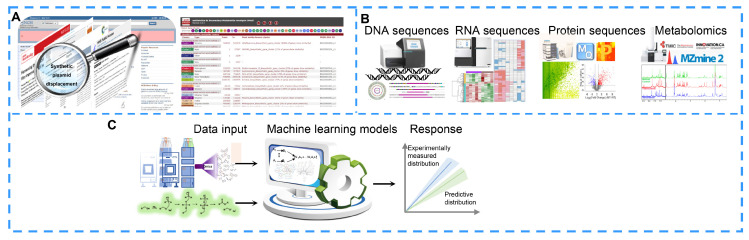
Bioinformatic analyses for natural product improvement. (**A**) Extracting the target information via online databases. (**B**) Discovering new natural products with the help of omics analyses. (**C**) Computer modeling predictions via metabolic networks.

**Figure 5 molecules-27-07480-f005:**
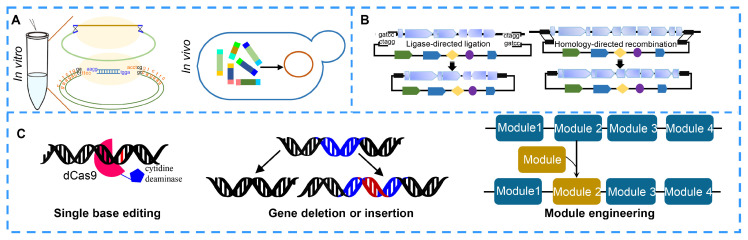
Tools for pathway reconstruction or engineering of natural product. (**A**) DNA assembly tools for assembling multiple DAN fragments of biosynthetic gene cluster in vivo or in vitro. (**B**) Capturing biosynthetic gene clusters of natural products by direct cloning. (**C**) Genome editing for improving natural products through single-base editing, gene deletion, gene insertion and module editing using CRISPR tools.

**Figure 6 molecules-27-07480-f006:**
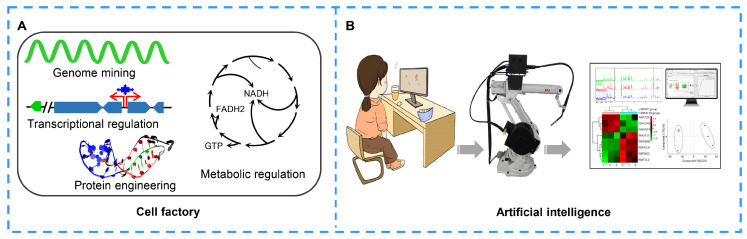
Biosystems for developing natural products. (**A**) Cell factory construction via genome mining, transcription regulation, metabolic engineering, protein engineering and transporter engineering. (**B**) Application of artificial intelligence in natural product improvement. Researcher writes programs and builds workstations, and robot accomplishes the instructions. In the last, researcher analyzes the data.

**Table 1 molecules-27-07480-t001:** All natural anticancer drugs approved from 1950 to 2021.

Active Ingredient	Drug Name	Target Cancer Type	Year Intro.	Source	Ref.
plant NPs					
asparaginase *Erwinia chrysanthemi*	Rylaze	acute lymphoblastic leukemia and lymphoblastic lymphoma	2021	*Erwinia chrysanthemi*	[31]
padeliporfin potassium	Tookad	prostate cancer	2015	a derivative of chlorophyll	[32]
paclitaxel injection concentrate for nanodispersion	PICN	metastatic breast cancer	2014	taxus species	[33]
homoharringtonine	Ceflatonin	myeloid leukemia	2012	*Cephalotaxus harringtonia*	[34]
ingenol mebutate	Picato	actinic keratosis	2012	*Euphorbia peplu*	[35]
cabazitaxel	Jevtana	castration-resistant metastatic prostate cancer	2010	a taxane derivative	[36]
vinflunine	Javlor	advanced and metastatic urothelial carcinoma	2010	a semisynthetic vinca-alkaloid	[37]
nanoparticle-based formulation of paclitaxel	Nanoxel	ovarian, non-small cell lung, breast, gastric, endometrial and pancreatic cancers	2007	taxus species	[38]
albumin-bound paclitaxel	Abraxane	ovarian, non-small cell lung, breast, gastric, endometrial and pancreatic cancers	2005	taxus species	[39,40]
belotecan HCL	Camtobell	small cell lung cancer	2004	a camptothecin analog	[41]
liposomal formulation of paclitaxel	Lipusu	non-small cell lung, breast, gastric, endometrial and pancreatic cancers	2003	taxus species	[42]
fulvestrant	Faslodex	advanced breast cancer	2002	a taxane plant product	[43]
arglabin	n.r.	oral squamous cell carcinoma, breast cancer	1999	*Artemisia* species	[44]
topotecan HCl	Hycamptin	small cell lung cancer	1996	a camptothecin analog	[45]
etoposide phosphate	Etopophos	advanced-stage Hodgkin lymphoma	1996	a semisynthetic derivative of podophyllotoxin	[46]
docetaxel	Taxotere	HER2-positive metastatic breast cancer	1995	yew tree	[47]
irinotecan HCl	Campto	colorectal and pancreatic cancer	1994	a camptothecin analog	[48]
paclitaxel	Taxol	non-small cell lung, breast, gastric, endometrial and pancreatic cancers	1993	pacific yew trees	[49]
masoprocol	Actinex	a potent sensitizer	1992	creosote bush	[50]
vinorelbine	Navelbine	non-small cell lung cancer	1989	a semisynthetic vinca alkaloid	[51]
solamargines	Curaderm	nasopharyngeal carcinoma cells	1989	*Solanum undatum*	[52]
elliptinium acetate	Celiptium	breast cancer	1983	a derivative of ellipticine derived from plant	[53]
etoposide	n.r.	extensive-stage small cell lung cancer	1980	a semisynthetic derivative of podophyllotoxin	[54]
vindesine	n.r.	Leukemia, non-small-cell lung cancer	1979	*Catharanthus roseus*	[55]
teniposide	n.r.	acute lymphoblastic leukemia	1967	a semisynthetic derivative of podophyllotoxin	[56]
vinblastine	n.r.	Hodgkin lymphoma	1965	*Catharanthus roseus*	[57]
vincristine	n.r.	hematologic malignancies and solid tumors	1963	nerium oleander	[57]
microbial NPs					
midostaurin	Rydapt	acute myeloid leukemia, advanced systemic mastocytosis	2017	a derivate of staurosporine produced by *Streptomyces staurosporeus*	[58]
romidepsin	Istodax	T cell lymphomas	2010	*Chromobacterium violaceum*	[59]
temsirolimus	Toricel	relapsed or refractory solid tumors	2007	a derivative of sirolimus produced by *Streptomyces hygroscopicus*	[60]
ixabepilone	Ixempra	breast cancer	2007	a derivate of epothilone B produced by *Sorangium cellulosum*	[61]
amrubicin HCl	Calsed	small-cell lung cancer	2002	an anthracyclin analogue	[62]
valrubicin	Valstar	non-muscle invasive bladder cancer	1999	a derivative of the anthracycline doxorubicin	[63]
zinostatin stimalamer	Smancs	hepatocellular carcinoma	1994	chemically synthesized by coupling one molecule of neocarzinostatin produced by *Streptomyces carzinostaticus*	[64]
idarubicin HCl	Zavedos	acute myelogenous leukemia	1990	an analogue of daunorubicin produced by *Streptomyces peucetius*	[65]
pirarubicin	Pinorubicin	lung cancer, breast cancer	1988	a novel anthracycline derivative of doxorubicin	[66]
epirubicin HCI	Farmorubicin	B cell lymphoma, head and neck cancer and other solid cancers	1984	a semisynthetic derivative of doxorubicin	[67]
aclarubicin	Aclacin	acute myeloid leukemia, hematologic cancers and solid tumors	1981	*Streptomyces galilacus*	[68]
peplomycin	Pepleo	cutaneous squamous cell carcinoma, prostatic cancer, breast cancer	1981	a derivative of bleomycin	[69]
asparaginase	n.r.	leukemia and lymphoma	1969	*Escherichia coli*	[70]
daunomycin	n.r.	acute promyelocytic	1967	*Streptomyces* species	[71]
bleomycin	n.r.	squamous cell carcinoma from head and necki, lymphomas, testicular carcinoma	1966	*Streptomyces verticillus*	[72]
doxorubicin	n.r.	ladder, breast, stomach, lung, ovaries, thyroid, soft tissue sarcoma	1966	*Streptomyces peucetius* var. caesius	[73]
actinomycin D	n.r.	solid tumors in children and choriocarcinoma in adult women	1964	*Streptomyces* species	[74]
mithramycin	n.r.	chronic and acute myeloid leukemia, testicular carcinoma	1961	*Streptomyces* species	[75]
mitomycin C	n.r.	bladder, breast carcinoma, head and neck malignancies, and some other gastrointestinal cancer	1956	*Streptomyces caespitosus*	[76]
leucovorin	n.r.	haematologic malignancies and osteosarcomas	1950	*Leuconostoc citrovorum*	[77]
antibody-conjugated NPs					
enfortumab vedotin-ejfv	Padcev	refractory bladder cancer	2019	a nectin-4-targeted antibody conjugated to monomethyl auristatin E, which is a toxin	[78]
polatuzumab vedotin	Polivy	relapsed or refractory diffuse large B-cell lymphoma	2019	an anti-CD79b antibody conjugated to monomethyl auristatin E, which is a toxin	[79]
trastuzumab emtansine	Kadcyla	HER2-positive early breast cancer	2019	a monoclonal antibody trastuzumaba conjugated to emtansine, which is a derivative of maytansine	[80]
inotuzumab ozogamicin	Mundesine	acute lymphoblastic leukemia	2017	an anti-22 antibody linked to calicheamicin	[81]
brentuximab vedotin	Adcetris	relapsed or refractory hodgkin lymphoma	2011	an anti-CD30 antibody conjugated to monomethyl auristatin E, which is a toxin	[82]
gemtuzumab ozogamicin	Mylotarg	acute myeloid leukemia	2000	anti-CD33 antibody linked to calicheamicin, a toxin	[83]
marine NPs					
lurbinectedin	Zepzelca	metastatic small cell lung cancer	2020	a marine-derived drug	[84]
aplidine	Aplidin	acute lymphoblastic leukemia	2018	*Aplidium albican*	[85]
trabectedin	Yondelis	soft tissue sarcomas—liposarcoma and leiomyosarcoma	2015	*Ecteinascidia turbinata*	[86]
eribulin	Halaven	breast cancer and soft-tissue sarcoma	2010	a macrocyclic ketone analogue of the halichondrin	[87]
hormones					
abiraterone acetate	Zytiga	castration-resistant prostate cancer	2011	a semisynthetic steroid	[88]
vapreotide acetate	Docrised	esophageal variceal bleeding	2004	a somatostatin analogue	[89]
exemestane	Aromasin	hormone receptor-positive breast cancer	1999	endocrine agent, steroidal compound	[90]
angiotensin II	Delivert	pancreatic cancer	1994	an endogenous hormone	[91]
formestane	Lentaron	breast cancer, prostatic cancer	1993	a steroid substrate analog	[92]
triptorelin	Decapeptyl	prostate cancer	1986	a decapeptide analog of luteinizing hormone releasing hormone	[93]
estramustine	n.r.	prostate cancer	1980	a stable estradiol	[94]
methyltestosterone	n.r.	breast cancer	1974	an anabolic–androgenic steroid	[95]
calusterone	n.r.	advanced breast cancer	1973	an androgenic steroid	[96]
megesterol acetate	n.r.	metastatic breast cancer	1971	n.r.	[97]
testolactone	n.r.	desmoid tumors, breast cancer	1969	a nonselective steroid	[98]
dromostanolone	n.r.	breast cancer	1961	an androgen steroid	[99]
nandrolone phenylpropionate	n.r.	hepatocellular adenomas	1959	a steroid	[100]
dexamethasone	n.r.	breast cancer, acute leukemia lymphoma	1958	a glucocorticoid	[101]
medroxyprogesterone acetate	n.r.	hormone-related cancers, cachexia syndrome	1958	a hormone progesterone variant	[102]
triamcinolone	n.r.	ocular conditions unresponsive to topical steroids	1958	a corticosteroid	[103]
methylprednisolone	n.r.	hodgkin lymphoma	1955	a corticosteroid	[104]
other NPs					
forodesine HCl	Mundesine	relapsed peripheral T cell lymphoma	2017	a purine nucleoside analogue	[105]
a nanoemulsion formulation containing 10% aminolaevulinic acid hydrochloride	Ameluz	lesion-directed and field-directed actinic keratosis	2012	the precursor of the endogenous photosensitizer Protoporphyrin IX	[106]
carfilzomib	Kyprolis	relapsed or refractory multiple myeloma	2012	a peptide epoxyketone	[107]
mifamurtide	Junovan	nonmetastatic osteosarcoma	2010	a conjugate of muramyl tripeptide linked to dipalmitoyl phosphatidyl ethanolamine	[108]
pralatrexate	Folotyn	relapsed/refractory peripheral T cell lymphomas	2009	a folic acid analogue	[109]
talaporfin sodium	Laserphyrin	esophageal cancer	2004	mono-L-asparthyl chlorine e6: NPe-6	[110]
methyl aminolaevulinate	Metvix	high-risk basal cell carcinoma	2001	the precursor of the endogenous photosensitizer Protoporphyrin IX	[111]
aminolevulinic acid	Levulan	premalignant and malignant diseases	2000	the first metabolite in the heme biosynthesis pathway	[112]
alitretinoin	Panretin	acute promyelocytic leukemia	1999	an endogenous vitamin A derivative, 9-cis-retinoic acid	[113]
cladribine	Leustatin	hairy-cell leukaemia	1993	a purine nucleoside analogue	[114]
cytarabine ocfosfate	Starsaid	acute myeloid leukemia	1993	an orally applicable prodrug of cytosine arabinoside	[115]
pentostatin	Nipent	lymphocytic leukemia	1992	an analogue of purine	[116]
mitobronitol	n.r.	acute myeloblastic leukemia	1979	n.r.	[117]

## Data Availability

Not applicable.

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
