# Peer review of "Breaking the Bottleneck in Anticancer Drug Development: Efficient Utilization of Synthetic Biology"

_molecules, 2022, doi:10.3390/molecules27217480_

Round 1
Reviewer 1 Report
This manuscript reports anticancer natural products approved in the last 70 years and their
mechanisms. The manuscript is well designed and informative. However, some modifications
are needed to publish, which are listed below. In my opinion, the manuscript can be accepted
for publication after major revisions.
Key issues:
1) Line 38: “However, … side effects.” Please cite some important references here.
2) Lines 78-80: The lines are less informative. So, I suggest the authors to rewrite this
section and shorten it.
3) Line 84: In general this is not true, for example, some forms of cancers are resistant
against some taxol or anthracycline drugs. So kindly modify the statement and cite
some recent important literature references.
4) Lines 103-104: please cite some references.
5) In table 1, the authors nicely categorise the different types of NPs, their active
ingredients and their year of approval. However, I found some problems to understand
the differences between some listed drugs. For example, in the very first and second
entries for “paclitaxel nanoparticles”, it is very hard to understand what kind of
differences they have. So, I would suggest the authors to add two more columns based
on their modifications and target cancer type. The entries are not following any order,
please arrange them year-wise.
6) Line 121: “NPs….structures” Kindly rewrite the line mentioning a clear statement,
instead of just mentioning “due to their various structures” may be author can use
electronic and geometrical structural variations.
7) Section 3.1: I encourage the authors to introduce the chemical structures of mentioned
drugs.
8) Line 177: Here, the authors can reproduce some graphical images of type II polyketide
pathways from some literature reports. This will enhance the clarity of the part.
9) Line 269: The author must cite some literature report here.
10) Section 4.3, last paragraph: author can discuss with the help of one or two examples
how microorganisms developed for the biosynthesis of a specific target. I strongly
believe that this will increase the clarity of the part.
11) Abbreviations: Please follow the alphabetical order.

Author Response
Dear Editors and Reviewers,
We thank all the editors and reviewers for the thoughtful comments. We have addressed all the questions and suggestions point-by-point as described below. In addition, we highlighted manuscript changes in red font in the text.
Reviewer # 1
Question 1. Line 38: “However, … side effects.” Please cite some important references here.
Answer1: We appreciate the reviewer’s suggestion. We cited two important references to support the chemotherapy-accompanied side effects on line 39 in the revised manuscript.
Question 2. Lines 78-80: The lines are less informative. So, I suggest the authors to rewrite this section and shorten it.
Answer2: We appreciate the reviewer’s suggestion. We shortened lines 78-80 and rewrote this section as “As a typical and general medical therapy for cancer, chemotherapeutic drugs can also damage normal cells and then cause a series of complications” on lines 78-79 in the revised manuscript.
Question 3. Line 84: In general this is not true, for example, some forms of cancers are resistant against some taxol or anthracycline drugs. So kindly modify the statement and cite some recent important literature references.
Answer 3: Thanks for the review’s comments. Actually, not all natural drugs avoid drug resistance, however, some of them prevent the development of drug resistance indeed. So, we modified the statement as “Different NPs, on the one hand, exhibit effective activities against various types of cancer, on the other hand, they play prominent roles in drug resistance via various mechanism” according to some recent important references on lines 82-84 in the revised manuscript.
Question 4. Lines 103-104: please cite some references.
Answer 4: We cited three recent references which are related to unapproved natural products with anticancer activity to support our statement on lines 103-104 in the revised manuscript.
Question 5. In table 1, the authors nicely categorise the different types of NPs, their active ingredients and their year of approval. However, I found some problems to understand the differences between some listed drugs. For example, in the very first and second entries for “paclitaxel nanoparticles”, it is very hard to understand what kind of differences they have. So, I would suggest the authors to add two more columns based on their modifications and target cancer type. The entries are not following any order, please arrange them year-wise.
Answer 5: Thanks for the review’s comments. We replaced these entries that are hard to understand with clearer statements and added a column based on the target cancer types of the natural products in the revised Table 1. We also arranged the entries year-wise in the same category and added the corresponding references to support these entries and in revised Table 1.
Question 6. Line 121: “NPs….structures” Kindly rewrite the line mentioning a clear statement, instead of just mentioning “due to their various structures” may be author can use electronic and geometrical structural variations.
Answer 6: We rewrote the sentence as “NPs suppress cancer development via diverse anticancer mechanisms due to their various chemical structures. For example, the intact lactone ring of camptothecins and the colchicine domain of podophyllotoxins are widely recognized as the active structures for their anticancer activities” on lines 122-124 in the revised manuscript.
Question 7. Section 3.1: I encourage the authors to introduce the chemical structures of mentioned drugs.
Answer 7: We introduced the chemical structures of mentioned drugs including plant-derived drugs, microorganism-derived drugs, natural product-based ADCs, marine natural products, hormones and other natural products. All these chemical structures were presented in the revised Figure 2.
Question 8. Line 177: Here, the authors can reproduce some graphical images of type II polyketide pathways from some literature reports. This will enhance the clarity of the part.
Answer 8: We appreciate the reviewer’s suggestion. We added a graphical image of type II polyketide pathway in the revised Figure 3C.
Question 9. Line 269: The author must cite some literature report here.
Answer 9: Thanks for the reviewer’s suggestion. We added some important references to support the statement on lines 279-280 in the revised manuscript.
Question 10. Section 4.3, last paragraph: author can discuss with the help of one or two examples how microorganisms developed for the biosynthesis of a specific target. I strongly believe that this will increase the clarity of the part.
Answer 10: We added two examples that how microorganisms developed for the biosynthesis of target compounds, including pyridinopyrone A, aloesaponarin II didesmethyl-mensacarcin and paclitaxel on lines 431-437 in the revised manuscript.
Question 11. Abbreviations: Please follow the alphabetical order.
Answer 11: Thanks for the reviewer’s suggestion. We arranged the abbreviations followed the alphabetical order in revised manuscript.

Reviewer 2 Report
1. What is the difference between the current review and this review? https://www.mdpi.com/2073-4409/9/9/2095/htm
2. Please change "cancer" as a keyword to "cancer therapy".
3. After this sentence "Thus, an increasing number of researchers are focusing on cancer research and therapy." there should be some reference." I suggest you add these review articles as references. The first one is for patent review for market anti-cancer drugs. The second one also is about triazoles as an easily prepared compound for anti-cancer products.
https://chemistry-europe.onlinelibrary.wiley.com/doi/abs/10.1002/slct.201902362
https://link.springer.com/article/10.1007/s11030-022-10406-8
4. The authors used "anticancer" and "anti-cancer". Please use "anticancer" or"anti-cancer" within the text.
5. If this review article is an updated review paper please add the word update in the title!
6. The authors used the abbreviation "MNPs" in "Conclusions and Future Perspectives" please write the complete full name as well.
7. Please write the years that this review is covering!
8. What is this title for table 1? "Table 1. This is a table. Tables should be placed in the main text near to the first time they are cited." I don't know what should I say! I should reject and resubmit this article only because of this mistake. How many mistakes are in the text? The authors should also provide some references for table 1.
Author Response
Dear Editors and Reviewers,
We thank all the editors and reviewers for the thoughtful comments. We have addressed all the questions and suggestions point-by-point as described below. In addition, we highlighted manuscript changes in red font in the text.
Reviewer # 2
Question 1. What is the difference between the current review and this review? https://www.mdpi.com/2073-4409/9/9/2095/htm
Answer 1: Thanks for the reviewer’s comment. Although both reviews focused on breaking the bottlenecks of cancer therapy, there are many differences. In this review (https://www.mdpi.com/2073-4409/9/9/2095/htm), authors concentrated on cellular immunotherapy (T-cell-receptor-based therapy) for cancer therapy instead of natural drugs. They discussed the unique potential of TCR therapy based on their advantages. Then, they summarized critical tasks of TCR based on neoantigen multitargeting and next-generation TCR according to T-Cell persistence, memory, fitness, tumor microenvironment and toxicity. Different from above review, we concentrated on natural drug therapies and the biosynthetic tools for drug discovery. We summarized these approved anticancer natural drugs (including plant-derived drugs, microorganism-derived drugs, marine drugs, antibody-conjugate drugs and hormones) from 1950 to 2021 and discussed their anticancer mechanisms. In our review, immunotherapy was a small section and it only included natural product-based antibody conjugates. More importantly, we discussed the currently advanced strategies and tools that were applied to discover those valuable natural products.
Question 2. Please change "cancer" as a keyword to "cancer therapy".
Answer 2: We change "cancer" as a keyword to "cancer therapy" on line 25 in the revised manuscript.
Question 3. After this sentence "Thus, an increasing number of researchers are focusing on cancer research and therapy." there should be some reference." I suggest you add these review articles as references. The first one is for patent review for market anti-cancer drugs. The second one also is about triazoles as an easily prepared compound for anti-cancer products.
https://chemistry-europe.onlinelibrary.wiley.com/doi/abs/10.1002/slct.201902362
https://link.springer.com/article/10.1007/s11030-022-10406-8
Answer 3: Thanks for the reviewer’s suggest. Both patent reviews are very interesting and important. So, we added these two references to support our statement on line 35 in the revised manuscript.
Question 4. The authors used "anticancer" and "anti-cancer". Please use "anticancer" or"anti-cancer" within the text.
Answer 4: Thanks for the reviewer’s suggest. We used "anticancer" within the revised manuscript.
Question 5. If this review article is an updated review paper please add the word update in the title!
Answer 5: Thanks for the reviewer’s comment. There are not many reports concentrating on discussing the mechanisms of all approved natural drugs and discussing the application of synthetic biology strategies and tools in development of clinical medicines at the same time. So, we think this review is not an updated review.
Question 6. The authors used the abbreviation "MNPs" in "Conclusions and Future Perspectives" please write the complete full name as well.
Answer 6: Thanks for the reviewer’s suggest. We replace "MNPs" with its complete full name "marine natural products" in the revised manuscript.
Question 7. Please write the years that this review is covering!
Answer 7: Thanks for the reviewer’s comments. We summarized the approved natural drugs covering from 1950 to 2021. The years that approved natural drugs is covering were mentioned in "abstract" and the second paragraph in "Clinical Applications of Natural Products in Cancer Therapy" section in previous manuscript ("FDA-approved Anticancer Natural Products" in the revised manuscript). We also emphasized the years that this review is covering in the last paragraph in "Introduction" section in the revised manuscript.
Question 8. What is this title for table 1? "Table 1. This is a table. Tables should be placed in the main text near to the first time they are cited." I don't know what should I say! I should reject and resubmit this article only because of this mistake. How many mistakes are in the text? The authors should also provide some references for table 1.
Answer 8: Thanks for the reviewer’s comments. We are very sorry for this mistake. When filling in the journal template to prepare the manuscript, table title was missed. We checked the whole revised manuscript carefully and corrected the title for table 1 in the revised manuscript. We also added corresponding references to support the statements of each entry in Table 1.
Round 2
Reviewer 1 Report
The authors have done all the recommended modifications meticulously. I am very happy to accept the updated manuscript for publication.
Thank you
Reviewer 2 Report
No more comments.